# A Visual Raman Nano−Delivery System Based on Thiophene Polymer for Microtumor Detection

**DOI:** 10.3390/pharmaceutics16050655

**Published:** 2024-05-14

**Authors:** Meng Li, Aoxiang Luo, Wei Xu, Haoze Wang, Yuanyuan Qiu, Zeyu Xiao, Kai Cui

**Affiliations:** 1College of Chemistry and Materials Science, Shanghai Normal University, Shanghai 200233, China; 1000513055@smail.shnu.edu.cn (M.L.); 1000513070@smail.shnu.edu.cn (H.W.); 2Department of Pharmacology and Chemical Biology, Translational Medicine Collaborative Innovation Center, Shanghai Jiao Tong University School of Medicine, Shanghai 200025, China; axluo2001@163.com (A.L.); 19821947682@163.com (W.X.); 1147109050@sjtu.edu.cn (Y.Q.)

**Keywords:** nano-delivery system, NIR excitation, resonant Raman scattering, microtumor imaging, in vivo pharmacokinetics

## Abstract

A visual Raman nano-delivery system (NS) is a widely used technique for the visualization and diagnosis of tumors and various biological processes. Thiophene-based organic polymers exhibit excellent biocompatibility, making them promising candidates for development as a visual Raman NS. However, materials based on thiophene face limitations due to their absorption spectra not matching with NIR (near-infrared) excitation light, which makes it difficult to achieve enhanced Raman properties and also introduces potential fluorescence interference. In this study, we introduce a donor–acceptor (D-A)-structured thiophene-based polymer, PBDB-T. Due to the D-A molecular modulation, PBDB-T exhibits a narrow bandgap of E_g_ = 2.63 eV and a red-shifted absorption spectrum, with the absorption edge extending into the NIR region. Upon optimal excitation with 785 nm light, it achieves ultra-strong pre-resonant Raman enhancement while avoiding fluorescence interference. As an intrinsically sensitive visual Raman NS for in vivo imaging, the PBDB-T NS enables the diagnosis of microtumor regions with dimensions of 0.5 mm × 0.9 mm, and also successfully diagnoses deeper tumor tissues, with an in vivo circulation half-life of 14.5 h. This research unveils the potential application of PBDB-T as a NIR excited visual Raman NS for microtumor diagnosis, introducing a new platform for the advancement of “Visualized Drug Delivery Systems”. Moreover, the aforementioned platform enables the development of a more diverse range of targeted visual drug delivery methods, which can be tailored to specific regions.

## 1. Introduction

A visual nano-delivery system (NS) is an extensively employed technique in biomedical imaging to offer crucial insights into physiological and pathological features, thus aiding in personalized diagnosis and treatment [1,2]. Nevertheless, existing visual NSs encounter challenges concerning photostability and biocompatibility when it comes to detecting microtumors and deep-seated tumors [3,4,5]. Fluorescence and Raman imaging, both of which possess superior spatial resolution and sensitivity, hold great promise for in vivo imaging applications in the field of biomedical imaging. Among them, fluorescence imaging is a widely employed technique [6,7]. Fluorescence imaging-based NSs have been developed in the research of cancer diagnosis and treatment [8]. It has been shown that poly (ethylene glycol)-block-polylactide (PEG-b-PLA) polymer micelles can be employed for optical imaging of tumors to monitor tumor growth and metastasis in real time by fluorescence imaging, thereby providing guidance for personalized treatment [9]. In addition, near-infrared fluorescence imaging utilizing a novel NIR fluorescent NS based on analogous proteins and PLA was employed for colon cancer detection. Thus, the visualization-guided delivery system can strongly guide tumor diagnosis [10]. However, fluorescent NSs often exhibit poor photostability and are prone to photobleaching [11]. Additionally, the emission spectra of fluorescence dyes typically have broad bands, leading to spectral overlap issues [12]. Meanwhile, Raman imaging is gaining increasing attention. It is an imaging modality associated with molecular vibrational modes, offering distinctive “fingerprint-like” spectra. Raman imaging provides spectra with extremely narrow peaks, enabling detection in complex biological environments [13,14]. Moreover, it possesses excellent resistance to photobleaching and photodegradation, making it suitable for long-term imaging applications [15,16,17].

Generally, spontaneous Raman scattering exhibits a small cross-section and low scattering efficiency, resulting in weak Raman signals. To achieve high-resolution Raman imaging, enhanced Raman scattering techniques are commonly utilized [18]. Among these techniques, surface-enhanced Raman scattering (SERS) is widely employed to enhance the scattering signal by adsorbing Raman-active molecules onto metal substrates such as gold (Au) [19,20]. However, the exceptional chemical stability and resistance to degradation and metabolism of Au and other metal substrates raise potential biocompatibility issues [21,22]. In contrast to SERS, organic resonance-enhanced Raman scattering does not require a metal substrate [23]. However, the signal intensity of resonance Raman scattering depends on the degree of matching between the absorption spectrum and the excitation light [24]. Currently, the absorption spectrum of a resonance Raman NS primarily lies within the Ultraviolet–Visible light range, and short-wavelength lasers (e.g., 514 nm, 532 nm) are commonly employed for resonance excitation [25,26]. Visible light is readily absorbed and scattered by tissues [27]. Conversely, NIR light exhibits enhanced penetration capabilities into biological tissues and reduces interference from autofluorescence, making it favorable for deep-tissue imaging [28]. Therefore, the development of a resonance Raman NS that can be excited by NIR light is urgently needed.

Polythiophene materials exhibit excellent biocompatibility and outstanding optical properties, rendering them valuable for the development of biological NSs. For example, Li Li et al. [29] prepared a class of fluorescence bio-probes based on thiophene or 3,4-ethylenedioxythiophene, which are specific for lipid droplet imaging, enabling the elucidation of differences in lipid droplets and cytoplasmic polarity. Wansu Zhang et al. [30] achieved high NIR spatial resolution fluorescence imaging probes by utilizing thiophene components of different lengths and connecting them with electron-withdrawing ester-substituted thieno[3,4-b]-thiophene (TT). Giada Onorato et al. [31] demonstrated optical control of a poly(3-hexylthiophene-2,5-diyl) (P3HT) NS for the regeneration and repair of tissue. Given the outstanding properties of thiophene-based polymers, our objective is to develop a visual Raman NS with excellent biocompatibility using multiple thiophenes as the structural components. However, thiophene-based polymers face the challenge of absorption spectra being far from the excitation light in the NIR range [32,33]. Therefore, it is essential to develop a thiophene-based polymer resonance-enhanced Raman NS that is compatible with NIR excitation light. As evidenced in the literature, the rational design of donor–acceptor (D-A) units in the main chain of polymers can regulate intra-molecular charge transfer between the donor and acceptor units, effectively red-shifting and extending the main absorption peak of the organic polymer NS into the NIR region [34]. This approach offers promising potential for the development of a thiophene-based polymer resonance-enhanced Raman NS with NIR excitation compatibility.

In this study, we selected the organic polymer PBDB-T (also known as PBDTBDD) containing the electron-withdrawing unit BDD with Di-thiophene structural components [35,36] and the electron-donating unit BDT with Tetra-thiophene structural components [37,38,39], as illustrated in the Figure 1 diagram. The investigation of the PBDB-T molecule revealed that following molecular modulation of the D-A type, the PBDB-T NS exhibited a significantly red-shifted absorption spectrum in comparison to the BDD NS. Furthermore, the absorption edge of the PBDB-T NS extended into the NIR region (700 nm~900 nm). Upon excitation with the commonly used 785 nm laser, the PBDB-T NS generated pre-resonance-enhanced Raman spectra, effectively avoiding strict resonance fluorescence interference, in contrast to the BDD NS [40,41,42]. This study highlighted the unique features of visual Raman NSs. By utilizing the characteristic Raman peak at 1425 cm^−1^ for biological imaging, the PBDB-T NS achieved an in vitro detection limit of 0.78 mg L^−1^, demonstrating substantial imaging potential in comparison to other reported resonance Raman NSs [43,44]. Upon intravenous administration, the PBDB-T NS accumulated in tumors through the enhanced permeability and retention (EPR) effect [45,46], facilitating the detection of microtumors measuring 0.5 mm × 0.9 mm, which demonstrated enhanced imaging capabilities when compared to the previously reported resonance Raman probes [44]. The PBDB-T NS also exhibited excellent biocompatibility due to its D-A-type molecular structure comprising thiophene components. In conclusion, the PBDB-T NS, which is modulated by D-A molecules, represents a promising NIR pre-resonance visual Raman NS for tumor diagnosis, offering a new platform for the development of “Visualized Drug Delivery Systems”.

### 1.1. Materials

All chemicals used in this article were purchased from commercial sources and used as is upon receipt. Poly[(2,6-(4,8-bis(5-(2-ethylhexyl)-thiophene-2-yl)-benzo[1,2-b:4,5-b′]-dithiophene))-alt-(5,5-(1′,3′-di-2-thienyl-5′,7′-bis(2-ethylhexyl)-benzo[1′,2′-c:4′,5′-c′]-dithiophene-4,8-dione)] (PBDB-T), Poly[benzo[1,2-c:4,5-c′]dithiophene-4,8-dione] (BDD) and the thiophene polymer poly(3-hexylthiophene-2,5-diyl) (P3HT) were purchased from Derthon Optoelectronics Materials Science Tech Co. (Shenzhen, China). 1,2-distearoyl-snglycero-3-1,2-distearoyl-snglycero-3-phosphoethanolamine-N[amino(polyethylene glycol)-2000] (DSPE-PEG) was purchased from AVT Pharmaceutical Technology Co., Ltd. (Shanghai, China). and sodium cholate was purchased from Bidepharm (Shanghai, China). Trichloromethane was supplied by Aladdin Biochemistry Co. (Shanghai, China).

### 1.2. Characterization

The absorption spectra of the nano-system were recorded using an Ultra-micro Spectro-photometer (BioDrop, Biochrom Inc., Cambridge, UK). An 800 μL aliquot of the PBDB-T NS solution was transferred into a quartz dish suitable for microscale solutions using a pipette gun and then placed into the sample detection chamber of the instrument. The instrument was configured to operate in absorbance spectroscopy mode, and the measurement process was completed accordingly. Spectral measurements were conducted five times, and representative spectra were selected for presentation.

The size of the nano-system was measured using a dynamic light scattering (DLS) analyzer (Zeta-sizer Nano ZS, Malvern, UK). The nanoparticle solution was appropriately diluted to 5 mg L^−1^ to ensure that the nanoparticles were uniformly dispersed and transferred to the DLS measurement cell. The assay was started in a Zeta-sizer Instrument according to the preset parameters, and at the end of the measurement, the particle size distribution of the nanoparticles, including the mean particle size and polydispersity index (PDI), was recorded.

The morphology of the NS was examined using a transmission electron microscope (TEM) (Talos L120C, Thermo Fisher Scientific, Waltham, MA, USA) at 120 kV. A 10 μL volume of the PBDB-T NS solution was deposited onto a formaldehyde-coated copper grid and allowed to air-dry for 30 min at room temperature. Subsequently, the sample was observed using transmission electron microscopy to characterize its morphology.

Raman detection and imaging were performed on a Renishaw inVia Raman microscope equipped with 532 nm, 785 nm, and 830 nm lasers, a 5× objective (NA 0.12; Leica, Wetzlar, Germany), and a 1020 × 256-pixel charge-coupled device detector. The output power of the lasers was measured using a handheld laser power meter (Edmund Optics Inc., Barrington, NJ, USA). And the instrumental limit of resolution was 1 cm^−1^.

The fluorescence spectrum was acquired using a multifunctional plate reader system (Thermo Fisher Scientific, USA), with suitable excitation wavelengths selected (532 nm, 600 nm, 633 nm, and 785 nm). A 96-well plate containing the PBDB-T NS at a concentration of 90 mg L^−1^ was employed. Spectral measurements were conducted five times, and representative spectra were carefully chosen for presentation.

Density Functional Theory (DFT) Calculation: Molecular optimization geometry (methyl substitution for side chain alkyl) of PBDB-T, BDD, and P3HT were calculated using Density Functional Theory (DFT) under Gaussian 16/B3LYP/6-31G(d) conditions. Version: Gaussian16; subversion: Revision A.03; release date: officially released to the world by Gaussian Inc., an American company, on 18 January 2017; the last update date is 31 August 2022.

The particle size distribution of the PBDB-T NS by TEM was simulated using Image J (Version 1.53t, 24 August 2022) [47].

### 1.3. Preparation of PBDB-T NS and BDD NS

DSPE-PEG (3 mg) and polymer (PBDB-T, BDD, or P3HT) (1.5 mg) were dissolved in 1 mL of trichloromethane to which 5 mL of sodium cholate solution (1% in water) was subsequently added. The mixed solution was sonicated for 3 min to produce an oil/water emulsion. The emulsion was then added to 30 mL of sodium cholate solution (0.1% in water) and stirred for 10 min. Finally, trichloromethane was removed by rotary evaporation for 30 min. The resulting nano-systems (NPs) were filtered and washed using an ultrafiltration tube (100 kDa) and then concentrated to a final concentration of 3 mg·mL^−1^. The concentration of NPs in aqueous solution is determined by the ratio of polymer mass to the volume of water (Figure 2).

### 1.4. Preparation of PBDB-T and BDD Film

Initially, the polymer (either PBDB-T or BDD) (0.3 mg) was dissolved in 1 mL of chloroform, after which the solution was drop-cast onto a quartz slide. The slide was then left in a fume hood for one day, allowing the solvent to completely evaporate before the measurement of the absorption spectrum.

### 1.5. In Vitro Raman Characterization

For the analysis of the characterization of PBDB-T and BDD solids, Raman measurements were carried out using a 5× objective lens, a 532 nm laser (power of 39.5 mW), a 785 nm laser (power of 84.5 × 10^−2^ mW), an 830 nm laser (power of 62.6 × 10^−2^ mW), an exposure time of 1 s, and a one-time accumulation setting. Spectral measurements were conducted five times, and representative spectra were selected for presentation.

To characterize the detection limit of the PBDB-T NS, we utilized a 785 nm laser and imaged using a 5× objective lens. Furthermore, the laser power was set to 84.5 mW, the acquisition time was set at 1 s, and only one accumulation was performed. Spectral measurements were conducted five times, and representative spectra were selected for presentation. The Raman spectra were analyzed utilizing the leveling baseline algorithm in WiRE4.3 software from Renishaw.

### 1.6. Cell Culture

CT26-Luc cells were provided by Imanis Life Sciences (Rochester, MN, USA) and then cultured in a culture environment at 37 °C with 5% CO_2_ using Roswell Park Memorial Institute-1640 (RPMI-1640) medium. An amount of 10% fetal bovine serum and 1% penicillin–streptomycin were mixed into RPMI-1640. Before preparation for the construction of the CT26-Luc in situ tumor model, the cells were cultured to reach confluency. 

### 1.7. Pharmacokinetics and Cytotoxicity of the PBDB-T NS

For pharmacokinetic studies, we used a BALB/c mouse model containing 5 mice per group and injected the PBDB-T NS via the tail vein at a dose of 20 mg kg^−1^. Blood sample collection time points were designed to be 1, 2, 4, 8, 20, 32, and 48 h post-injection, and 50 μL of blood samples were collected each time. After the blood sample was centrifuged, the supernatant was removed and placed on a slide, and the Raman signal was measured with the use of Raman spectroscopy and labeled as a percentage (%) of the relative injected dose. Subsequently, pharmacokinetic parameters were calculated using a one-compartment open model. Cell viability of the PBDB-T NS on the CT26-Luc cells was evaluated using the Cell Counting Kit-8 (CCK-8) assay following the guidelines of the manufacturer, Beyotime (Nantong, China). First, 5 × 10^4^ CT26-Luc cells per milliliter were placed in 96-well microwell plates in RPMI-1640 medium containing 10% FBS. Twenty-four hours after cell attachment, 12 and 24 h incubations were performed using RPMI-1640 medium containing 0–250 mg L^−1^ of PBDB-T NS (5 replicate sets for each treatment). Afterward, the PBDB-T NS were removed and cells were washed three times with PBS. An amount of 10 μL of CCK-8 dye and 100 μL of RPMI-1640 were added to each well and then incubated for 2 h. The plates were analyzed using an enzyme marker (Molecular Devices SpectraMax M2e) analysis device. Absorbance readings were taken at 450 nm and the value read was proportional to the number of live cells, while cell viability was expressed as a percentage of absorbance relative to untreated controls.

### 1.8. Animal Studies

In compliance with the study design approved by the Animal Protection Committee, all animal experiments were conducted at the Animal Resource Center of Shanghai Jiao Tong University School of Medicine. BALB/c nude mice, approximately 6 weeks old, used for animal imaging studies were purchased from Shanghai SLAC Laboratory Animal Co., Ltd. (Shanghai, China). The mice were anesthetized by 2% isoflurane inhalation during the imaging experiments.

### 1.9. Establishment of Subcutaneous or Orthotopic CT26-Luc Colon Tumor Model

Based on a previously reported method [48], BALB/c mice were used for a subcutaneous tumor model with CT26-Luc cells (~2 × 10^6^), which were injected subcutaneously into the right side of each mouse. To construct the in situ CT26-Luc colon tumor model, the abdominal skin of mice was first sterilized using 0.5% (*w*/*v*) iodophor solution. Next, a small central incision (about 3~5 mm) was cut through the skin of the lower abdomen to expose the cecum. Subsequently, CT26-Luc cells (approximately 2 × 10^6^) suspended in 15 μL of PBS were injected into the cecum wall. Upon completion of the procedure, the injected cecum was placed back into the abdominal cavity and the wound was closed using biodegradable sutures. To monitor the growth of colon tumor lesions, an IVIS spectral imaging system was used to detect tumors by intraperitoneal injection of D-luciferin (15 mg/mL, 200 μL), thereby measuring bioluminescent signals.

### 1.10. In Vivo Raman Imaging

In the tumor Raman imaging study, BALB/c nude mice were first injected with PBS liquid (10 mg kg^−1^) containing the PBDB-T NS. Then, mice with subcutaneous CT26-Luc colon tumor models were scanned 24 h later, and Raman signals were collected at 1425 cm^−1^ for Raman imaging. In this procedure, a 5× objective lens, a 785 nm laser excitation source, a power of 84.5 mW (power density: 3.5 × 10^2^ w cm^−2^), an exposure time of 0.52 s, and a single accumulation in the point-and-scan acquisition mode were used, with an integration time of 43 min. For mice in the in situ CT26-Luc colon tumor model, the abdominal skin was incised under anesthesia to expose the cecum, and then Raman scanning was performed. The Raman imaging parameters utilized consisted of a 5× objective lens, 785 nm laser excitation at 84.5 mW power, a 0.3 s exposure time, collection of Raman signals at 1425 cm^−1^, and a single cumulative shot in streamline-scan imaging acquisition mode, followed by an integration time of 42 min.

### 1.11. Histologic Analysis

Tumors were identified and excised under Raman imaging, and tumor samples were embedded in an optical cutting temperature compound (OCT) (Sakura Finetek Inc., Tokyo, Japan) for frozen sectioning. After frozen sectioning, tissue sections of 4 μm thickness were used for routine H&E staining. This staining method is often used to observe and evaluate the cellular morphology and organization of tissues.

### 1.12. Data Processing

The quantitative results were presented as means ± S.D. (standard deviation). Student’s *t*-tests were employed to assess inter-group differences. Statistical significance was denoted as * for *p* < 0.05, ** for *p* < 0.01, *** for *p* < 0.001, and **** for *p* < 0.0001.

## 2. Results and Discussion

### 2.1. Preparation and Spectral Characterization of the PBDB-T NS and BDDNS

We first investigated the Raman spectroscopic properties of PBDB-T and BDD (Figure 1A,B) in their solid-state forms. PBDB-T is a thiophene-based polymer with a D-A structure unit. As shown in Figure 1C–E, the Raman peaks of PBDB-T in the wavenumber range of 1070 cm^−1^ and 1540 cm^−1^ exhibited the highest intensities when excited by a 785 nm laser, compared to excitation with 532 nm and 830 nm lasers. Therefore, subsequent studies were mainly conducted using 785 nm excitation. Then, the molecular optimization geometry (methyl substitution for side chain alkyl) of PBDB-T was calculated using Density Functional Theory (DFT) under RB3LYP/6-31G(d) conditions for Raman vibration mode. 

Under laser excitation, PBDB-T exhibits enhanced characteristic Raman peaks within the range of 800 cm^−1^ to 1600 cm^−1^ as follows: The peak at 1425 cm^−1^ corresponds to the benzene ring C=C stretching mode and thiophene C=C stretching mode in the BDT unit, as well as the C=C stretching and C-H rocking modes of the bridged thiophene between BDD and BDT; the peak at 1460 cm^−1^ belongs to the benzene ring C=C stretching mode and thiophene C=C stretching mode in the BDT unit, as well as the C-H rocking mode of the bridged thiophene; the peak at 1480 cm^−1^ is attributed to the thiophene C=C stretching mode in the BDT unit; the peak at 1540 cm^−1^ corresponds to the benzene ring C=C stretching mode, thiophene C-H rocking mode in the BDT unit, as well as the bridged thiophene C-H rocking mode; and finally, the peak at 1070 cm^−1^ is associated with the C-H rocking mode of the BDD unit and the bridging thiophene C-H rocking mode. Among these Raman peaks, the peak at 1425 cm^−1^ showed the highest intensity and was selected as the main peak for further experiments. In contrast to PBDB-T, the BDD polymer did not exhibit enhanced Raman spectra when excited with 532 nm, 785 nm, or 830 nm lasers in the solid state. To transform PBDB-T into a Raman imaging NS with long circulation capability in vivo, PBDB-T NS with an average size of 10.2 nm were prepared by mixing PBDB-T polymer with DSPE-PEG_2000_ using a single emulsion–solvent evaporation method [49], and transmission electron microscopy (TEM) images confirmed the spherical morphology of the NSs (Figure 1F) (simulated size by TEM: 9.509 nm).

In conclusion, the D-A molecular structure modulates the intra-molecular charge distribution of the PBDB-T NS, thereby enhancing excellent Raman spectral properties. This has inspired further investigations into the mechanisms underlying Raman enhancement and the potential applications of in vivo Raman imaging.

### 2.2. Donor–Acceptor(D-A) Molecular Regulation Mechanism of the PBDB-T NS for Raman Enhancement

To further investigate the inspired Raman performance of PBDB-T over BDD, we initially compared their absorption spectra and energy gap variations. First, BDD nanoparticles were prepared using the aforementioned method, as depicted in Figure 2A, with a pale yellow color (2.5 mg L^−1^). 

The absorbance spectra of the film and NS states exhibited a range of approximately 220–400 nm, indicating that the 785 nm excitation was significantly outside the absorption spectrum of the BDD NS, as shown in Figure 2C. The PBDB-T NS appeared deep blue (2.5 mg L^−1^), with strong absorption observed in the UV–Vis–NIR spectra in the range of approximately 500~700 nm for both the film and NS, displaying two distinct absorption peaks (Figure 2B). The peak at 578 nm was attributed to the π-π* transition of the main chain’s conjugated backbone, while the 622 nm peak was attributed to the interchain π-π* transition induced by the π-π stacking of the main chain [50]. Both the film and NS states of PBDB-T exhibited similar absorption peaks, indicating pronounced aggregation in both states. 

The position of the 785 nm excitation was 70 nm from the edge of the absorption peak (i.e., 715 nm), which is outside the PBDB-T NS absorption spectrum but close enough to induce enhanced Raman scattering through 785 nm laser pre-resonance excitation (Figure 2C). Molecular optimization geometry (methyl substitution for side chain alkyl) of PBDB-T, BDD, and P3HT were calculated using Density Functional Theory (DFT) under Gaussian 16/B3LYP/6-31G(d) conditions. The highest occupied molecular orbitals (HOMOs) and lowest unoccupied molecular orbitals (LUMOs) were plotted (Figure 2D). It was observed that the electronic distribution in the LUMO energy orbitals of PBDB-T is mainly located on the electron-withdrawing unit side, while in the HOMO energy orbitals, the electronic distribution is mainly on the electron-donating unit side. This results in a significantly smaller bandgap (E_g_) compared to BDD, which explains the absorption red-shift of the PBDB-T NS relative to the BDD NS. Furthermore, a polythiophene polymer P3HT NS was prepared using the aforementioned method, with an absorption spectrum ranging from approximately 350 nm to 655 nm and an absorption peak at 515 nm, resulting in a blue-shift relative to the absorption spectrum of the PBDB-T NS (Figure 2E) due to P3HT’s larger bandgap (E_g_) compared to PBDB-T (Figure 2F).

The matching degree of the excitation light and the absorption spectrum play a crucial role in the enhancement of resonance Raman scattering for organic materials [51]. When the absorption spectrum of a polymer coincides with the excitation light, strict resonance Raman scattering occurs, resulting in signal enhancement but also a strong likelihood of intense fluorescence interference, ultimately leading to a sharp decrease in the detected Raman scattering signal [52]. Conversely, under non-resonant conditions, although fluorescence interference is greatly reduced due to the excitation light being far from the absorption spectrum of the organic material, the collected Raman signal also rapidly diminishes. Thus, by adjusting the molecule structure, it is possible to tune the absorption edge to be near the excitation light, which can reduce fluorescence interference while enhancing Raman scattering signals [53,54]. This represents a promising approach to Raman enhancement known as “pre-resonance Raman enhancement”.

In this study, the PBDB-T polymer, which has been molecularly tuned through the BDD electron-accepting unit and the BDT electron-donating unit, leads to the redistribution of electrons from the original non-interacting orbitals of the acceptor and donor into the hybridized orbitals of PBDB-T, resulting in the creation of new HOMO and LUMO energy levels after polymerization (Figure 2D). The new polymer HOMO orbitals are typically distributed on the donor unit, while LUMO orbitals are typically distributed on the acceptor unit, and a strong electron donor can raise the energy levels of the HOMO orbitals [55,56]. As shown in Figure 2D, molecular tuning of the D-A structure results in an increase in the HOMO energy level of the PBDB-T polymer, leading to a reduced bandgap, and causing the absorption edge to shift towards the NIR (Figure 3A). Therefore, the PBDB-T NS will exhibit different Raman properties under excitation by 532 nm laser and NIR lasers (785 nm and 830 nm). The position of the 785 nm excitation light near the absorption edge of the PBDB-T NS allows for enhanced Raman scattering signals through pre-resonance excitation, reducing fluorescence interference (Figure 3C). Although the 830 nm excitation light is far from the 622 nm absorption peak of the PBDB-T NS and outside of its absorption spectrum, it effectively reduces fluorescence interference but results in a very low collected Raman scattering signal (Figure 3D,E). Ultimately, the Raman signal intensity obtained with 785 nm excitation at 1425 cm^−1^ is approximately 13.6 times higher than that obtained with 830 nm excitation (Figure 3F). On the other hand, the 532 nm excitation light, which is close to the 578 nm absorption peak of PBDB-T NS and within its absorption spectrum (Figure 3A), results in strict resonance excitation, leading to both resonant enhanced Raman signals and intense fluorescence interference (Figure 3B,E,G).

Consequently, the Raman signal intensity detected at 1425 cm^−1^ under 532 nm excitation is only approximately one-third of that obtained under 785 nm excitation (Figure 3F). Regarding the BDD NS, with absorption peaks at 275 nm and an absorption edge at around 500 nm (Figure 3A), both 532 nm, 785 nm, and 830 nm are far from their absorption peaks, resulting in the inability to obtain resonant-enhanced Raman signals (Figure 3B–D). It is worth noting that the positions of the Raman spectral peaks of the PBDB-T NS are identical to those obtained in the solid state (Figure 1D and Figure 3C). Furthermore, even at a low concentration of 2 mg L^−1^, the Raman signal at 1425 cm^−1^ can still be detected for the PBDB-T NS, with a calculated detection limit of 0.78 mg L^−1^ (Figure 3H,I), indicating a high Raman detection sensitivity of PBDB-T NS.

Under excitation at 532 nm and 633 nm where the laser wavelengths coincide with the absorption peaks of PBDB-T, strict resonance excitation occurs (Figure 4A), resulting in intense fluorescence signals with emission spectra ranging from approximately 650 nm to 850 nm (Figure 4B). This corresponds to the fluorescence interference generated by PBDB-T probes under resonant excitation conditions during Raman detection, which leads to lower Raman enhancement properties under resonant excitation, as depicted in Figure 3E,F.

Conversely, no significant fluorescence signal was observed under excitation at 785 nm (Figure 4C), indicating the absence of fluorescence interference during near-infrared excitation. As illustrated in Figure 3E,F, under 785 nm laser excitation, the PBDB-T probes achieve optimal Raman enhancement properties by attaining pre-resonance Raman enhancement effects while simultaneously avoiding fluorescence interference. However, under 830 nm laser excitation, which is further away from the absorption peak, although the possibility of fluorescence interference is further reduced, the resonance enhancement effect for Raman is also diminished, resulting in the weakest Raman enhancement properties observed under 830 nm laser excitation. Finally, we conducted additional measurements of the fluorescence spectra of BDD NPs under 532 nm laser excitation, as shown in Figure 4A,D, and found no detectable fluorescence signal under 532 nm laser excitation. Therefore, the lack of Raman enhancement in BDD probes under 532 nm laser excitation (Figure 3B) is not due to fluorescence signal interference, but rather because the absorption spectrum of BDD probes is too far from the 532 nm excitation light (Figure 4A).

In conclusion, the enhancement of pre-resonance Raman signals depends on the relative distance between the absorption edge of the polymer NS and the excitation light (785 nm), which is 70 nm, with the Raman signal intensity gradually decreasing as the excitation light (830 nm) moves away from the absorption edge. At the same time, pre-resonance Raman also avoids fluorescence interference caused by strict resonance. Therefore, following molecular tuning of the D-A type, the absorption spectrum of the PBDB-T NS is optimally matched with the 785 nm laser, resulting in the best enhancement of Raman properties. This positions the PBDB-T NS as a promising candidate for high-resolution in vivo Raman imaging.

### 2.3. Stability, Cytotoxicity, In Vivo Pharmacokinetics of the PBDB-T NS

To investigate the photobleaching resistance, the PBDB-T NS was excited in PBS solution using a 785 nm laser for 60 s to complete Raman stability measurement, indicating the excellent photostability of the PBDB-T Raman NS relative to the fluorescent NS (Figure 5A,B). Furthermore, the particle size of the prepared PBDB-T NS showed no significant change even after storage at room temperature for over 5 months, indicating the excellent storage stability of the PBDB-T NS (Figure 5C). To assess the stability of the Raman property of the PBDB-T NS in physiological environments, the PBDB-T NS was cultured in FBS for 48 h, and Raman spectra were obtained at different time points, demonstrating the maintenance of good Raman stability of the PBDB-T NS in serum (Figure 5D,E). Inspired by the excellent Raman properties of the PBDB-T NS under 785 nm excitation at 1425 cm^−1^, we explored the cytotoxicity and in vivo pharmacokinetics of the PBDB-T NS. The cytotoxicity of the PBDB-T NS was evaluated by analyzing the cell viability of CT-26 cells at different concentrations (0~250 mg L^−1^) using CCK-8 (Cell Counting Kit-8). As shown in Figure 5F, even after co-incubation of the NS with the cells for 12 and 24 h, it was observed that the cell viability remained above 80% even at an NS concentration as high as 250 mg L^−1^, indicating the low toxicity of the PBDB-T NS to cells.

To evaluate the pharmacokinetics, the PBDB-T NS was injected intravenously at a concentration of 20 mg L^−1^, and blood samples were taken from the ophthalmic vein at different time points. Quantitative analysis of the Raman signal intensity at 1425 cm^−1^ in the centrifuged serum was performed to follow the distribution of the PBDB-T NS from the bloodstream to the tissues. As shown in Figure 5G,H, as the PBDB-T NS gradually distributed from the blood to the systemic tissues, the concentration of the NS in the blood slowly decreased, with an in vivo circulation half-life of 14.5 h [57]. These results lay the foundation for subsequent imaging of colon tumors using the PBDB-T NS.

### 2.4. In Vivo Intraoperative Raman Imaging of Tumors and Metastatic Microtumors, and In Vivo Non-Invasive Raman Imaging of Deep Tumor Tissue

Visualization of small tumors (~4 mm) during surgery, especially metastatic tiny tumors (~3 mm) that are often missed because they are difficult to observe, is critical to improving cancer outcomes and survival [58]. Therefore, the detection of microtumors in surgery has emerged as a promising direction for Raman imaging. To explore whether the PBDB-T NS can achieve this goal in the process of Raman detection, we established an in situ CT 26-Luc colon tumor model based on a previously reported method and administered the PBDB-T NS to mice via intravenous injection (10 mg L^−1^). Due to the EPR effect, the PBDB-T NS passively accumulated in the tumor region. Characteristic Raman signals at 1425 cm^−1^ were detected at the site of in situ tumors 24 h post-administration (Figure 6A,B), and positive areas of the characteristic Raman signal were confirmed as tumor regions by hematoxylin and eosin (H&E) staining (Figure 6C). Additionally, Raman imaging of the PBDB-T NS effectively detected a metastatic microtumor measuring 0.5 mm × 0.9 mm (Figure 5A Merged), which is crucial for comprehensive and complete tumor clearance. These results demonstrate that the PBDB-T NS can accurately distinguish in situ tumor tissue, metastatic microtumors, and normal tissue through Raman imaging.

Non-invasive in vivo imaging of deep tissues plays a significant role in tumor diagnosis and treatment [59]. Based on existing methods, we established a subcutaneous CT 26 colorectal tumor model and utilized the accumulation of the PBDB-T NS in the subcutaneous tumor for deep tissue Raman imaging. Following intravenous administration of the PBDB-T NS (10 mg L^−1^), Raman imaging was conducted 24 h later. In comparison with adjacent normal tissue, it was found that the NS could detect the tumor in deep tissues (Figure 7A,B). This was further confirmed by hematoxylin and eosin (H&E) staining analysis, which revealed that the positive areas of the characteristic Raman signal corresponded to the tumor region (Figure 7C). Additionally, Raman imaging of tumors in the non-dosed control group by PBDB-T NPs, as depicted in Figure 7D,E, further elucidated the passive tumor-targeting aggregation of the probe for tumor imaging diagnosis.

After the completion of tumor Raman imaging, the main organs of the tumor-bearing mice were removed and subjected to histological evaluation. In comparison with the control group, no significant tissue abnormalities were observed in the major organs including the heart, liver, spleen, lungs, and kidneys (Figure 8). Additionally, an analysis of mouse blood chemistry and hematology was conducted (Table 1). This indicates that the PBDB-T NS is a biologically safe in vivo Raman NS.

Visual NSs have the full potential to deliver drugs, increase drug targeting, and clarify drug metabolism kinetics [60]. For example, a fluorescence-based prodrug NS allows tumor-specific stimulation [61], while some researchers have developed an NS based on NIR-II fluorescence imaging that can be used to monitor responsive drug release and tumor targeting [62]. Based on this, our pre-resonance visual Raman NS is expected to guide therapy along with tumor diagnosis in the future.

## 3. Conclusions

In this study, we have successfully explored the application of an organic polymer, PBDB-T, with a donor (Tetra-thiophene structural components) and acceptor (Di-thiophene structural components) as a visual Raman NS. PBDB-T exhibited a red-shifted absorption spectrum close to the NIR excitation light. When using 785 nm excitation light for optimal matching excitation, the PBDB-T NS achieved enhanced resonant Raman spectra without fluorescence interference. Therefore, by collecting Raman signals at 1425 cm^−1^ after intravenous administration, a high-resolution diagnosis of microtumors measuring 0.5 mm × 0.9 mm was achieved through Raman imaging. Additionally, it allowed for the bioimaging of deep tumor tissues. Our study unveils the potential of PBDB-T as a promising NIR pre-resonance visual Raman NS for tumor diagnosis, offering a new pathway for the development of polymer Raman nano-systems incorporating multiple thiophenes as structural components, and providing a modifiable foundational platform for future visual drug NS development.

## Data Availability

Data are contained within the article.

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
