# Peer review of "A Visual Raman Nano−Delivery System Based on Thiophene Polymer for Microtumor Detection"

_pharmaceutics, 2024, doi:10.3390/pharmaceutics16050655_

Round 1

Reviewer 1 Report (New Reviewer)

Comments and Suggestions for Authors

The research paper entitled "A Visual Raman Nano-Delivery System based on Thiophene 2 Polymer for Microtumor Detection" is both innovative and promising. The authors address a critical challenge in Raman nano-delivery systems by introducing a donor-acceptor (D-A) structured thiophene-based polymer, PBDB-T. By carefully modulating the molecular structure, PBDB-T achieves a narrow bandgap and a red-shifted absorption spectrum, extending into the near-infrared (NIR) region. This strategic design allows for optimal excitation with 785 nm light, resulting in ultra-strong pre-resonant Raman enhancement while avoiding fluorescence interference.

My only question is why the peak at 1425 cm-1 does not shift between the different wavelengths. My experience leads me to observe peak shifts of the same molecule as the lasers used vary. The authors should comment on this phenomenon, after which the paper can be accepted as is.

Comments on the Quality of English Language

Nothing

Author Response

Reviewer 2 Report (New Reviewer)

Comments and Suggestions for Authors

The manuscript by Li and colleagues provides significant information on microtumor detection. The current version has several minor errors, but the reference section needs a deep revision to meet the journal's requirements. Due to the high number of expected corrections, the referee suggests a major revision.

- The phrase "Visual Raman Nano-Delivery System" appears in the title. Why did the authors feel the need to repeat it in the keywords?

- In Chapter 2.2, the authors failed to inform the reader about the wavelength used in the DLS experiments. The particle size calculation depends on the wavelength used, and it is essential to communicate it. It is unclear whether fluorescence interfered with the DLS experiments.

- The chemical compound P3HT is not defined. It is unclear whether it is identical to orP3HT mentioned in line 150 and the flowchart. This inconsistency also causes a mismatch between the caption text and the content of the flowchart (top left stamp).

- Low contrast between black characters and dark grey background makes the flowchart difficult to understand.

The authors have the TEM image of PBDB-T NS, which is also suitable for determining the particle size distribution. They did not perform this calculation. There is a lot of free software available for this purpose. Usually, the DLS and TEM images give different particle size distributions.

- The homo/lumo energies were calculated using a quantum chemical package. The citation of the software should follow the requirements of the developers (please refer to the help file or the vendor's homepage for the correct referencing). The exact identification of the software package has been included, including the version, subversion, release date, last update date, and homepage.    

- The references need reformatting to eliminate duplication of reference numbers.

Journal abbreviations should include dots.

Most journal names are in capital letters, while others are not. The reviewer's knowledge suggests that journal names should follow some conventions.

Some journal names are not abbreviated.

Reference 27 is formatted incorrectly, and the year appears twice.

Comments on the Quality of English Language

Spellcheck is mandatory.

Author Response

Reviewer 3 Report (New Reviewer)

Comments and Suggestions for Authors

The manuscript pharmaceutics-2951360 "A Visual Raman Nano-Delivery System based on Thiophene Polymer for Microtumor Detection" by Li et al. describes the synthesis of novel thiophene-based polymer, the study of spectral properties and iv vivo biomedical applications. I think this paper will be of interest to the readers of Pharmaceutics.

However, I have a few questions and comments:

1) Some methods (TEM, DFT, DLS, UV-Vis, fluorescence spectroscopy, etc.) are not described in "Materials and Methods". Please add a detailed description of how the experiment was conducted for each method used in this study. No information about most of the devices is also available.

2) The authors performed DLS experiments. How did the nanoparticle size and polydispersity index (PDI) change (before and after 5 months)? Please add the instrumental images obtained in the original program.

3) Since the obtained systems are planned to be injected into living organisms, it is necessary to evaluate not only cytotoxicity but also hemotoxicity.

4) I recommend comparing the results obtained by the authors with previous results obtained by other scientific groups.

Round 2

Reviewer 2 Report (New Reviewer)

Comments and Suggestions for Authors

The authors made the necessary corrections and clarified the referee's concerns. Now, it is suitable for publication.

Author Response

Reviewer 3 Report (New Reviewer)

Comments and Suggestions for Authors

I thank the authors for answering my questions and improving the manuscript.

Please correct "Wavalength" to "Wavelength" in the Figures.

Author Response

This manuscript is a resubmission of an earlier submission. The following is a list of the peer review reports and author responses from that submission.

Round 1

Reviewer 1 Report

Comments and Suggestions for Authors

The authors present an interesting study into the design, optimisation and application of a thiophene-based organic material, PBDB-T, as a unique Raman probe for bioimaging applications. The authors present a some interesting structure-absorption features of the thiophene materials and the justification for biological imaging in the NIR window. In mouse models, the authors present convincing data to support the localisation of the probe in the tumour environment. The authors could expand the discussion as to the specificity of tumour targeting – is the localisation of the probe specific to the tumour or is a systemic localisation achieved? The authors also demonstrate the biocompatibility of their probe in cell culture and in the mouse model but fail to include the Raman imaging results in the organs tested (in Fig. 7) which may shed some further insight into the localisation of the PBDB-T probe. That being said, the manuscript is clearly written and presented to a high standard. I have some minor comments for the authors to consider.

Page 2 line 62: repetition of the Li et al.

In Figure 1 C-E the authors present the baseline corrected Raman spectra of the thiophene systems at different wavelengths. Could the authors explain why the baseline is not at (or near) 0 a.u. following baseline correction and why the intensity values of the baselines differ so much in the BDD spectra? Could the authors comment on whether any enhancement in the BDD spectra would be expected at 532 nm, given the absorption profile presented in Scheme 1? Is the signal swamped by background fluorescence?

Line 271: First of all, using the above method, BDD NPs were also prepared, as depicted in Figure 2A, presenting a pale yellow (2.5 mg/L). This sentence is incomplete.

Line 428: The authors comment on the “Characteristic Raman signals at 1425 cm-1 were detected at the site of the in-situ tumor 24 hours post-administration”. What is used to target the probe to the tumour site? Is the probe systemically localised within the mouse? The Raman imaging neatly demonstrates the localisation of in the tumour, but it is not clear if this is a specific accumulation.

Line 463: The authors conclude that “…that PBDB-T NP is a biologically safe in vivo Raman imaging probe” yet fail to present the Raman imaging data to support their claim i.e. the Raman imaging of the tissues in Fig 7 is missing. Could the authors explain this discrepancy? Is their probe localised systemically rather than specifically in the tumour mass?

Comments on the Quality of English Language

Very minor comment as noted above.

Reviewer 2 Report

Comments and Suggestions for Authors

Meng Li and colleagues used thiophene-based Raman probes, namely PBDB-T, for in vivo imaging. The PBDB-T-Trobe exhibited a energy bandgap of 2.63 eV and a red-shifted absorption spectra, with the absorption edge extending into the near-infrared (NIR) region. PBDB-T NPs serve as a very sensitive Raman probe for in vivo imaging, allowing for the detection of microtumor areas of 0.5 mm×0.9 mm. Additionally, they effectively diagnose tumor tissues located at greater depths. The work is quite interesting, although it needs some revisions before to being published.

1.    First of all, the author must clarify the scheme in order to better comprehend the reader.

2.    The PBDB-T probes' emission spectra and a discussion of the NIR impact should be included.

3.    Details about the Raman band must be included or added to the table independently.

4.    The author should provide statistical information on the Raman measurement, such as the number of spectra recorded each time.

5.    Figure 6: The author need to provide data from a control experiment using Raman imaging.

6.    For measurement, a Raman laser spot size must also be added.

Reviewer 3 Report

Comments and Suggestions for Authors

Dear Editor,

I am sharing my review of the pharmaceutics-2860197entitled: Thiophene polymer Raman probe PBDB-T based on NIR optimal matching excitation for Microtumor imaging.

The study discusses the limitations of using thiophene-based organic polymers for Raman imaging due to their absorption spectra not matching with NIR excitation light. A donor-acceptor structured thiophene-based polymer, PBDB-T, was introduced to overcome this limitation. PBDB-T exhibits a narrow bandgap and a red-shifted absorption spectrum, extending the absorption edge into the NIR region. Upon optimal excitation with 785 nm light, PBDB-T achieves ultra-strong pre-resonant Raman enhancement while avoiding fluorescence interference. As an intrinsically sensitive Raman probe for in vivo imaging, PBDB-T NPs enable the diagnosis of microtumor regions and successfully diagnose deeper tumor tissues, providing a new pathway for developing thiophene-based polymer Raman probes. The paper requires minor revision. Please fix the following issues:

Introduction

The scheme and the text should be transferred to the result section

M/M

A graphical representation of the whole workflow would be good.

In vivo Raman imaging:

·         Please add information concerning analyzed range (800 to 1600 cm-1), spectral resolution, a lateral resolution (field of view), additions of measurement points and an integration time.

·         Data Processing (Software, data processing -> baselin correction and or normalization)

Histologic analysis

Which Tumor classification was used? Which Microscope was used?

Results and Discussion:

Indicate the tumor region in Figure 5 and Figure 6.

Figure 5A: there are no blue arrows in the Figure -> please fix it.

Figure 5B: what is 1 2 3 in the figure -> explain it in the caption

Figure 6B: what is 1 2 (what regions tumor non tumor ?) in the figure -> explain it in the caption

The work is exciting!

Best